# Macrophage Activation Assays to Evaluate the Immunostimulatory Capacity of *Avibacterium paragallinarum* in A Multivalent Poultry Vaccine

**DOI:** 10.3390/vaccines8040671

**Published:** 2020-11-10

**Authors:** Robin H. G. A. van den Biggelaar, Willem van Eden, Victor P. M. G. Rutten, Christine A. Jansen

**Affiliations:** 1Department of Biomolecular Health Sciences, Division of Infectious Diseases and Immunology, Faculty of Veterinary Medicine, Utrecht University, Yalelaan 1, 3584CL Utrecht, The Netherlands; r.h.g.a.vandenbiggelaar@uu.nl (R.H.G.A.v.d.B.); w.vaneden@uu.nl (W.v.E.); v.rutten@uu.nl (V.P.M.G.R.); 2Department of Veterinary Tropical Diseases, Faculty of Veterinary Science, University of Pretoria, Private bag X20, Hatfield 0028, South Africa

**Keywords:** inactivated poultry vaccine, *Avibacterium paragallinarum*, infectious coryza, potency test, in vitro, HD11 cell line, nitric oxide, cytokines, lipopolysaccharides

## Abstract

High-quality vaccines are crucial to prevent infectious disease outbreaks in the poultry industry. In vivo vaccination tests are routinely used to test poultry vaccines for their potency, i.e., their capacity to induce protection against the targeted diseases. A better understanding of how poultry vaccines activate immune cells will facilitate the replacement of in vivo potency tests for in vitro assays. Using the chicken macrophage-like HD11 cell line as a model to evaluate innate immune responses, the current explorative study addresses the immunostimulatory capacity of an inactivated multivalent vaccine for infectious bronchitis, Newcastle disease, egg-drop syndrome, and infectious coryza. The vaccine stimulated HD11 cells to produce nitric oxide and to express pro-inflammatory cytokines *IL-1β*, *TNF,* and *IL-12p40,* chemokines *CXCLi1* and *CXCLi2*, and the anti-inflammatory cytokine *IL-10*, but only when inactivated *Avibacterium paragallinarum*, the causative agent of infectious coryza, was present. Lipopolysaccharides from *Avibacterium paragallinarum* were crucial for the production of nitric oxide and expression of *IL-1β* and *CXCLi1*. The described immune parameters demonstrate the capacity of this multivalent vaccine to activate innate immune cells and may in the future, combined with antigen quantification methods, contribute to vaccine quality testing in vitro, hence the replacement of current in vivo vaccination tests.

## 1. Introduction

In the poultry industry, chickens are routinely vaccinated against infectious diseases to maintain flock health. Vaccine batches are subjected to routine quality control (QC) to ensure their potency, i.e., their capacity to induce protection against the targeted infectious diseases. For many poultry vaccines, current QC test approaches include in vivo vaccination-challenge or vaccination-serology tests, which require large numbers of chickens. Alternatively, poultry vaccines could be tested in accordance with the consistency approach [1], which implies that quality can be proven by in vitro methods that demonstrate the similarity between the product profiles of a new batch and reference batches of proven clinical efficacy and safety. The characteristics that determine the potency of a vaccine have to be predetermined and may include properties like antigen quantity and the capacity to stimulate innate immune responses [1,2,3].

Infectious coryza is an upper respiratory tract infection caused by the Gram-negative bacterium *Avibacterium (Av.) paragallinarum*, which is responsible for worldwide economic losses due to mortality and reduced egg production in chickens [4]. Breeding and laying hens can be protected against infectious coryza by inactivated vaccines comprising multiple *Av. paragallinarum* isolates of serovars A, B, and C [5]. Like the vaccine used in this study, *Av. paragallinarum* can be incorporated in inactivated multivalent vaccines that also comprise infectious bronchitis virus (IBV), egg-drop syndrome ’76 virus (EDSV) and Newcastle disease virus (NDV), to reduce the time and costs of vaccination [6]. Inactivated NDV vaccines can be assessed for potency using an in vitro enzyme-linked immunosorbent assay (ELISA) [7,8,9] and inactivated IBV and EDSV vaccines can be assessed for potency by in vivo vaccination-serology tests [10,11], which have a lower impact on animal welfare than traditional vaccination-challenge tests. In contrast, inactivated infectious coryza vaccines still rely on vaccination-challenge tests to prove their potency and are hence in need of reliable in vitro alternatives [12,13]. The capacity of a vaccine to induce protective immunity depends in part on its capacity to stimulate innate immune cells through recognition of immunostimulatory constituents including adjuvants and intrinsic pathogen-associated molecular patterns (PAMPs) like bacterial cell wall components [14,15]. The importance of PAMPs for effective vaccination has already been demonstrated for human vaccines against bacterial pathogens [16] like *Bordetella pertussis* [17,18] and *Mycobacterium tuberculosis* [14,19]. However, these immunostimulatory constituents have not been described for most poultry vaccines. Here, we investigated whether infectious coryza vaccines contain PAMPs that activate innate immune cells.

In general, bacterial PAMPs are recognized by innate immune cells through pattern recognition receptors (PRRs), which are broadly expressed by phagocytes like macrophages. Furthermore, the importance of macrophages in detecting bacterial PAMPs during vaccination has been demonstrated in macrophage-depletion studies in mice [20], which justifies the use of this cell type in in vitro vaccine quality assessment. A prime candidate to characterize innate immune responses evoked by poultry vaccines is the chicken macrophage-like cell line HD11. The HD11 cell line was originally described as LSCC-HD (MC/MA1) [21] but later renamed to HD11 [22]. This cell line recognizes bacteria using a broad repertoire of PRRs, including most chicken toll-like receptors (TLRs) [23] and mannose receptors. Activation of HD11 cells can be observed by increased phagocytosis [24,25,26], cytokine expression [27,28], and nitric oxide production by the inducible nitric oxide synthase (iNOS) [27,28,29,30,31]. As vaccines for *Av. paragallinarum* are commonly multivalent and also used to protect chickens against other pathogens, we used two multivalent vaccines in the current study. To identify which responses were directed against *Av. paragallinarum*, an octavalent vaccine comprising inactivated IBV, NDV, EDSV, and 5 strains of *Av. paragallinarum* was used next to a trivalent vaccine that only contained inactivated IBV, NDV, and EDSV. In this explorative study, we aimed to identify candidate immune parameters for an in vitro QC test for inactivated poultry vaccines against *Av. paragallinarum* and contribute to the replacement of current in vivo vaccination-challenge tests.

## 2. Materials and Methods

### 2.1. HD11 Cell Culture and Stimulation

Batches of chicken macrophage-like HD11 cells [21] were suspended in complete Roswell Park Memorial Institute (RPMI)-1640 medium with 50% FBS (both from Gibco, Life Technologies Limited, Paisley, UK) and 10% DMSO (Honeywell, Bucharest, Romania) and stored at −140 °C. After thawing, they were maintained and propagated in a complete RPMI-1640 cell culture medium supplemented with GlutaMAX-I, phenol red, HEPES, 10% FBS, 200 U/mL penicillin, and 200 U/mL streptomycin (all from Gibco) at 37 °C, 5% CO2, in Corning 75 cm^2^ cell culture flasks (Sigma-Aldrich, Saint Louis, MO, USA). The cells were passaged twice weekly by washing the cells in Dulbecco’s phosphate-buffered saline without calcium and magnesium (PBS; Lonza, Basel, Switzerland) and detaching adherent cells using a 0.25% trypsin/EDTA solution supplemented with phenol red (Gibco). For the experiments, HD11 cells were harvested after 3 to 20 passages using the trypsin/EDTA solution, counted, and resuspended at a concentration of 200,000 cells/mL in complete RPMI-1640 medium. The cells were seeded at 1 mL/well in Corning Costar 24-well cell culture plates (Sigma-Aldrich) and cultured overnight at 37 °C and 5% CO_2_.

HD11 cells were exposed to various stimuli to assess their activation using nitric oxide production as a readout. First of all, TLR agonists were given at eleven different concentrations with increments of 10^0.25^ between consecutive concentrations to create dose-response curves and to test the ability of HD11 cells to recognize different PAMPs. Stimuli included 1.0–300 ng/mL lipopolysaccharides (LPS) from *Escherichia* (*E.*) *coli* O127:B8 (Sigma-Aldrich) to target TLR4, 0.2–60 ng/mL Pam3CSK4 to target the TLR2/1 heterodimer, 0.2–60 μg/mL zymosan from *Saccharomyces* (*S.*) *cerevisiae* to target the TLR2/6 heterodimer, 0.1–30 μg/mL high molecular weight (1.5-8kb) polyinosinic-polycytidylic acid (poly[I:C]) to target TLR3, 0.1–30 μg/mL resiquimod (R848) to target TLR7, and 3–1000 ng/mL CpG oligonucleotides (ODN) 2006 to target TLR21 (all InvivoGen, San Diego, CA, USA). In addition, HD11 cells were exposed to established tri- an octavalent inactivated poultry vaccines kindly provided by Boehringer Ingelheim (Ingelheim am Rhein, Germany) as a partner of the VAC2VAC consortium (http://www.vac2vac.eu/). The octavalent vaccine comprised of whole-inactivated IBV, NDV, EDSV, and inactivated isolates of five serovars of *Av. paragallinarum*, including one isolate of serovar A, three of serovar B, and one of serovar C. The trivalent vaccine used in this study comprised whole-inactivated IBV, NDV, and EDSV only. Both vaccines were adjuvanted as mineral oil-based water-in-oil emulsions. The vaccines were prepared such that a single chicken vaccination dose corresponded to 0.5 mL vaccine. HD11 cells were exposed to the vaccines at a fixed concentration of 1 µL/mL or to different concentrations ranging from 0.2 to 30 µL/mL to create dose-response curves.

In accordance with a previous publication [7], total antigenic fractions were extracted from the vaccines by adding vaccine to isopropyl myristate (Sigma-Aldrich) in a 1:5 ratio, vigorous mixing on a vortex mixer for 1 min, and centrifugation at 1000× *g*. The lower water phase contained the antigenic fraction, which was gently resuspended and collected. HD11 cells were exposed to extracted antigens at a fixed concentration of 0.5 µL/mL or different concentrations ranging from 0.2–30 µL/mL to create dose-response curves. Furthermore, *Av. paragallinarum* bacteria were purified from 200 µL extracted antigens by centrifuging three times at 15,000 *g* and washing the bacterial pellet in PBS. Finally, the bacterial pellet was resuspended in 200 µL PBS. A pellet was absent when the trivalent vaccine comprising only viral antigens was exposed to the same procedure. HD11 cells were exposed to the purified bacteria at different concentrations ranging from 0.1–10 µL/mL to create dose-response curves.

The contribution of LPS, present within the antigens extracted from the octavalent vaccine, to the activation of HD11 cells was determined using the LPS-binding antibiotic polymyxin B (Sigma-Aldrich) [32]. For these experiments, 300 ng/mL *E. coli* LPS, 300 ng/mL CpG ODN2006, 0.5 µL/mL octavalent vaccine and 1.0 µL/mL extracted antigens were resuspended in complete RPMI-1640 and pre-incubated with 1, 10, or 100 µg/mL polymyxin B for 1 h at 37 °C. Subsequently, these mixtures were added to the HD11 cells and incubated for 48 h to evaluate the production of nitric oxide or for different periods of time between 0–48 h to evaluate the gene expression of candidate biomarkers over time.

### 2.2. Griess Test to Evaluate Nitric Oxide Production by Stimulated HD11 Cells

Nitric oxide production by HD11 cells was measured after 48 h of stimulation by the Griess test as previously described [33]. Griess test reagent was made by dissolving N-(1-naphtyl)ethylenediamine at 3 g/L and sulfanilamide at 10 g/L (both from Sigma-Aldrich) in 2.5% phosphoric acid (Supelco, Merck, St. Louis, MO, USA) and mixing the two solutions 1:1. From the stimulated HD11 cell cultures, 50 μL supernatant was harvested, transferred to a 96-well flat-bottom plate (Corning B.V. Life Sciences, Amsterdam, the Netherlands), and mixed with 50 μL/well Griess test reagent. The mixture turns purple upon reaction with nitrite ions in the cell culture supernatant. The median optical density at 540 nm was determined for each well using a FLUOstar Omega microplate reader (BMG Labtech, Ortenberg, Germany). The corresponding nitrite concentrations were calculated according to a nitrite standard curve made from a dilution series between 3.13–200 μM sodium nitrite (Sigma-Aldrich). For HD11 stimulated with TLR agonists or purified inactivated *Av. paragallinarum* bacteria, the results of the Griess test were plotted using the four-parameter logistic regression model from GraphPad Prism 8 software (GraphPad Software)
(1)$$E\left(C\right)=E_{\max}+\frac{E_{\max}-E_{\min}}{1\;+{\left(\frac{C}{{\mathrm{EC}}_{50}}\right)}^{n}},$$
where E(C) = effect at a given concentration (C), E_max_ = maximum observed effect, E_min_ = minimum observed effect, EC_50_ = half-maximum effective concentration, and n = Hill coefficient. The model was used to determine the maximum nitric oxide production (E_max_) and the concentration that gives half-maximal nitric oxide production (EC_50_) as readouts for the capacity of the stimuli to induce nitric oxide production.

### 2.3. Relative Expression of iNOS and Cytokines Using Real-Time Quantitative PCR

To determine iNOS and cytokine gene expression, HD11 cells were harvested, either 8 h after stimulation or at different time points between 0–48 h (as indicated in Figure 3), using 200 µL PBS + 5 mM UltraPure EDTA (Invitrogen™, Life Technologies Europe BV, Bleiswijk, the Netherlands) and centrifugated at 400× *g*. The supernatant was discarded, and the pelleted cells were lysed in RLT buffer (Qiagen GmbH, Hilden, Deutschland) and stored at −20 °C until further processing. After thawing, RNA isolation was performed with the RNeasy Mini Kit (Qiagen GmbH, Hilden, Deutschland) including a DNase treatment using the RNase-Free DNase Set (Qiagen GmbH, Hilden, Deutschland). Next, cDNA was prepared using the iScript cDNA Synthesis Kit (Bio-Rad Laboratories B.V., Veenendaal, the Netherlands) according to the manufacturer’s instructions. RT-qPCR reactions were performed with 100 nM FAM-TAMRA-labelled TaqMan probes (listed in Table 1) together with 600 nM primers and TaqMan Universal PCR Master Mix or without probes using 400 nM primers and SYBR Green Master Mix (all from Applied Biosystems, Life Technologies Europe BV, Bleiswijk, the Netherlands). RT-qPCR reactions were performed using a CFX Connect qPCR detection system (Bio-Rad), using a program of 40 cycles with melting temperatures of 58 (SYBR Green probes) or 59 °C (TaqMan probes), and analyzed with CFX Maestro software (Bio-Rad). All RT-qPCR reactions were evaluated for proper amplification efficiency (95–105%) using serial dilutions of reference cDNA or plasmids before testing the samples (data not shown). RT-qPCR reactions were performed in triplicate for every sample. Changes in gene expression over time upon stimulation were assessed using t = 0 h as a reference time point and expressed as 2^-ΔΔCt^-values, according to the Livak method [34] with Ct being the number of cycles before a signal above the threshold (background) level was reached. The results were normalized to gene expression levels of the housekeeping genes *28S* and *GAPDH*. An exception is the *IL-10* gene expression data of Appendix A, which were expressed as 40-Ct values according to a previous publication [35] as this method was more suitable when one or more data points did not reach a signal above the threshold within 40 cycles.

### 2.4. Flow Cytometric Assessment of HD11 Cell Viability after Stimulation

HD11 cells were harvested 48 h after stimulation using 200 µL PBS + 5 mM UltraPure EDTA and centrifuged at 400× *g*. The cells were transferred to a 96-well V-bottom plate, washed in PBS with 0.5% bovine serum albumin, 0.005% sodium azide (FACS buffer; both from Sigma-Aldrich), and stained for viability in 200 µL FACS buffer with 1.25 µg/mL 7-aminoactinomycin D (7-AAD; BD Biosciences, Pharmingen, San Diego, CA, USA). The staining of the cells was analyzed using the 488 nm laser of a CytoFLEX LX flow cytometer (Beckman Coulter Inc., Brea, CA, USA). The fraction of viable 7-AAD-negative HD11 cells was determined using FlowJo Software v. 10.5 (FlowJo LCC, Ashland, OR, USA). The cytotoxicity of the extracted antigens and purified bacteria from the octavalent vaccine was expressed as the concentration at which 50% of HD11 cells had died, i.e., the 50% lethal concentration (LC_50_).

### 2.5. Statistical Analysis

Statistical analyses were performed using GraphPad Prism 8 software (GraphPad Software Inc., San Diego, CA, USA). When the assumptions of normally distributed data and residuals were met, a one-way ANOVA with Holm–Sidak’s multiple comparisons test was used to test for statistically significant differences between stimulated and unstimulated control samples. Relative gene expression data were log-transformed to create normally distributed data. When the assumptions of normality were not met, a non-parametric Kruskal–Wallis test with Dunn’s multiple comparisons test was used instead. A *p*-value of <0.05 was considered statistically significant.

## 3. Results

### 3.1. The Inactivated Octavalent Poultry Vaccine Induces HD11 Cells to Produce Nitric Oxide

HD11 cells were stimulated for 48 h by different TLR agonists derived from or mimicking components from bacteria (Pam3CSK4, LPS, CpG), viruses (R848, CpG, poly[I:C]), and fungi (zymosan). Nitric oxide production was analyzed using the Griess test (Figure 1). The most potent inducer of nitric oxide production was Pam3CSK4 (EC_50_ = 23.0 ng/mL; E_max_ = 101 μM nitrite), which stimulates the chicken TLR2/1 heterodimer. Other potent stimuli were the TLR4 agonist *E. coli* LPS (EC_50_ = 122 ng/mL; E_max_ = 93.5 μM nitrite) and the TLR21 agonist CpG ODN2006 (EC_50_ = 127 ng/mL; E_max_ = 94.7 μM nitrite). High concentrations of R848 (EC_50_ = 6.42 μg/mL; E_max_ = 57.1 μM nitrite), stimulating TLR7, and *S. cerevisiae* zymosan (EC_50_ = 3.68 μg/mL; E_max_ = 66.3 μM nitrite), stimulating both the TLR2/6 heterodimer and Dectin-1, were required to induce nitric oxide production. Nitric oxide was not detected upon exposure to high molecular weight poly(I:C) oligonucleotides. Overall, these results demonstrate that nitric oxide production can be used as a readout to determine the capacity of various compounds to stimulate innate immune cells like macrophages.

Next, we investigated whether an inactivated octavalent poultry vaccine for IBV, NDV, EDSV, and five serovars of *Av. paragallinarum* contains any immunostimulatory constituents that may stimulate nitric oxide production by HD11 cells. Stimulation with this vaccine for 48 h at concentrations ranging from 0.56 to 3.2 μL/mL induced significantly higher nitric oxide production compared to unstimulated cells (E_max_ = 34.7 µM nitrite at 1.0 μL/mL) (Figure 2a). At doses > 1.0 μL/mL, nitric oxide production gradually decreased, which can be explained by the observed cytotoxicity of the vaccine at high concentrations, as assessed by 7-AAD viability staining (Appendix A).

Gene expression of *iNOS* was measured to investigate the activation of the nitric oxide production pathway over time as the accumulation of nitric oxide cannot be detected at early time points (*not shown*). Expression of *iNOS* was strongly upregulated between 4 and 6 h after exposure to 1.0 μL/mL octavalent vaccine, reaching a 59-fold increase at t = 6 h in comparison to t = 0 h (Figure 2b). The maximum increase in *iNOS* expression was observed at t = 24 h (311-fold) and expression remained high at least up to 48 h (289-fold), which was the last time point assessed.

### 3.2. Stimulation with the Octavalent Vaccine Results in Enhanced Gene Expression of Cytokines and Chemokines

Cytokine gene expression by vaccine-stimulated HD11 cells was followed over time. As early as 1 h after exposure to the vaccine, HD11 cells upregulated gene expression levels of the pro-inflammatory cytokine *IL-1β* (4.3-fold) and IL-8-like chemokines *CXCLi1* (2.9-fold) and *CXCLi2* (3.3-fold) (Figure 3a–c), which increased up to 8 h after exposure to respectively 584-, 99.9-, and 225-fold. Expression of *IL-1β* subsequently decreased to 30.0-fold at 24 h and 9.3-fold at 48 h as compared to t = 0 h (Figure 3a), whereas expression of *CXCLi1* and *CXCLi2* remained high up to 48 h (Figure 3b,c). HD11 cells showed a modest upregulation of the pro-inflammatory cytokine *TNF* after exposure to the octavalent vaccine, reaching its peak at t = 8 h with a 4.0-fold increase in expression (Figure 3d). In contrast, stimulation with the vaccine did not induce gene expression of *IL-6* and *IFN-α* at any of the time points studied (Figure 3e,f). Expression of the anti-inflammatory cytokine *IL-10* was elevated after 6 h to 19.5-fold and remained at that level up to 48 h (Figure 3g). Expression of the Th1 or 17-inducing cytokine *IL-12p40*, depending on its heterodimerization with respectively IL-12p35 [36] or IL-23p19 [37], was increased maximally 11.7-fold at t = 48 h (Figure 3h). Expression of Th1-inducing cytokine *IFN-γ* was found slightly decreased at 24 and 48 h (4.50- and 4.14-fold decrease respectively) compared to t = 0 h (Figure 3i). The Th2-inducing cytokine *IL-4* and Th1-inducing cytokine *IL-12p35* were not detected (*not shown*). Taken together, HD11 cells showed increased expression of pro-inflammatory cytokines *IL-1β* and TNF, and chemokines *CXCLi1* and *CXCLi2* within 8 h after exposure to the octavalent vaccine, which was followed by the induction of the anti-inflammatory cytokine *IL-10* and the Th1/Th17-inducing cytokine *IL-12p40*.

### 3.3. Av. Paragallinarum Antigens Contribute to the Stimulatory Capacity of the Octavalent Vaccine

To investigate whether *Av. paragallinarum* antigens were able to stimulate HD11 cells, the responses induced by the octavalent vaccine were compared to responses induced by a trivalent vaccine containing the same viral antigens but without bacterial *Av. paragallinarum* antigens. In contrast to the octavalent vaccine, the trivalent vaccine did not lead to nitric oxide production by HD11 cells (Figure 4a). These results strongly suggested that the *Av. paragallinarum* antigens were responsible for the nitric oxide production by HD11 cells. Next, antigens were extracted from the emulsion vaccines using isopropyl myristate. As shown in Figure 4b, the antigens extracted from the octavalent vaccine induced more nitric oxide production than the octavalent vaccine itself (50.2 µM vs. 6.7 µM nitrite at 0.3 μL/mL, respectively). In contrast, the antigens extracted from the trivalent vaccine did not induce nitric oxide production. Compared to the complete octavalent vaccine with an LC_50_ of 18 μL/mL, the extracted antigens were more cytotoxic for HD11 cells with a LC_50_ of 0.26 μL/mL (Appendix A). To determine whether these cytotoxic effects were due to the bacterial antigens or other vaccine constituents, the bacteria were recovered from the antigenic fraction of the octavalent vaccine by centrifugation at high speed, followed by extensive washing in PBS, and resuspended in PBS according to the original volume of antigenic fraction (Appendix A). The bacteria stimulated HD11 cells to produce high concentrations of nitric oxide (E_max_ = 57.1 µM nitrite at 1.0 μL/mL) over a wide range of concentrations (Figure 4c), whereas the cytotoxicity of the purified bacteria (LC_50_ = 4.2 μL/mL) was reduced compared to the antigenic fraction (LC_50_ = 0.26 μL/mL) (Appendix A). The potency of the purified bacteria to induce nitric oxide production, as expressed by an EC_50_-value, was found to be = 0.276 µL/mL.

Next, the gene expression levels of *iNOS* and cytokines were determined 8 h after stimulation with either the tri- or octavalent vaccine. Exposure to the octavalent vaccine led to significantly increased expression of *iNOS* (36.4-fold), *IL-1β* (71.4-fold), *TNF* (1.7-fold), *CXCLi1* (46.9-fold), *CXCLi2* (55.7-fold) (Figure 4e–i), and *IL-10* (40-Ct: 4.0 for octavalent vaccine vs. 1.1 for unstimulated; Appendix A). In contrast, the trivalent vaccine did not induce the expression of *iNOS* or any of the cytokines. Expression *IFN-α* was slightly decreased after stimulation with either the tri- (2.1-fold decrease) or octavalent (2.0-fold decrease) vaccine (Figure 4j).

### 3.4. Activation of HD11 Cells by the Octavalent Vaccine Largely Depends on Av. Paragallinarum Antigen-Associated LPS

Finally, the antigenic fraction of the octavalent vaccine was pre-incubated with the LPS-binding antibiotic polymyxin B to determine the contribution of LPS, embedded in the cell wall of *Av. paragallinarum* bacteria, to HD11 cell activation. First, *E. coli* LPS (Figure 5a) and CpG ODN2006 (Figure 5b), acting as positive and negative controls, were both preincubated with polymyxin B for 1 h and subsequently administered to HD11 cell cultures. Polymyxin B significantly inhibited LPS-induced nitric oxide production (36.5% reduction), whereas CpG-induced nitric oxide production remained unchanged. Next, the antigenic fraction of the octavalent vaccine was preincubated with polymyxin B and administered to the cells (Figure 5c), which resulted in a significant reduction in nitric oxide production of 65.7% as compared to exposure to the antigenic fraction alone. Moreover, the treatment of the antigenic fraction of the octavalent vaccine with polymyxin B resulted in a significant reduction in the expression of *iNOS* (67.6% reduction; Figure 5d), *IL-1β* (80.0% reduction; Figure 5e), and *CXCLi1* (58.9% reduction; Figure 5f). The expression levels of *CXCLi2* (35.1% reduction; Figure 5g) and *IL-10* (43.5% reduction; Figure 5h) tended to be inhibited by pre-incubation with polymyxin B, but this was not statistically significant. Taken together, these experiments suggest that LPS present in *Av. paragallinarum* antigens significantly contributed to the activation of HD11 cells upon exposure to the octavalent vaccine.

## 4. Discussion

In this explorative study, we aimed to investigate the capacity of an octavalent vaccine containing inactivated IBV, NDV, EDSV, and five serovars of *Av. paragallinarum* to activate innate immune cells, in view of identifying potential immune parameters that could be used in the future for in vitro vaccine QC testing of inactivated *Av. paragallinarum*. The vaccine was found to activate the chicken macrophage-like cell line HD11 shortly after stimulation, resulting in the expression of the nitric oxide-producing enzyme *iNOS* and the pro-inflammatory cytokines *IL-1β*, *TNF*, *CXCLi1,* and *CXCLi2*. From 6 h after stimulation, HD11 cells expressed the anti-inflammatory cytokine *IL-10*, which inhibits TLR signaling [38] and may be responsible for the subsequent return of *IL-1β* and *TNF* expression to baseline levels. In contrast, HD11 cells were not activated by a trivalent vaccine containing the same inactivated viral antigens but without inactivated *Av. paragallinarum* antigens, indicating that bacterial PAMPs were important for the evoked immune response. Furthermore, pre-treating the extracted antigens with the LPS-neutralizing antibiotic polymyxin B [32] largely averted nitric oxide production and cytokine expression by HD11 cells, showing that bacterial LPS was an important immunostimulatory factor of the octavalent vaccine. Hence, stimulation of HD11 cells by the octavalent vaccine must at least partly be dependent on TLR4, the designated PRR for LPS [39]. This is supported by an in vivo study, showing upregulation of the TLR4 signaling pathway in the nasal tissues of chickens infected with *Av. paragallinarum* [40].

The differences in macrophage activation by the octa- and trivalent vaccine were striking and may be explained by the presence or absence of bacterial PAMPs from *Av. paragallinarum* acting as an endogenous adjuvant of the octavalent vaccine. This is in agreement with in vivo data demonstrating that chickens developed strong granulomatous reactions involving macrophages at injection sites of inactivated bacterial *Av. paragallinarum* or *Mycoplasma gallisepticum* vaccines, but not at injection sites of inactivated viral NDV, IBV, avian reovirus, or infectious bursal disease virus vaccines [41,42,43]. The presence of bacterial antigens containing PAMPs in the multivalent vaccine most likely also boosts the immune responses against the antigens of viral origin, as demonstrated for an experimental vaccine against both inactivated viral influenza A and *Streptococcus pneumonia* in mice [15].

The role of nitric oxide production and cytokines to the induction of vaccine-mediated protection against *Av. paragallinarum* in chickens is still unknown. A genome-wide association study in chickens has shown that there is an association between serological responses and small nucleotide polymorphism within and surrounding the gene encoding for iNOS, suggesting that iNOS may also be important in chickens during vaccination [44]. Studies with iNOS, IL-17, IL-4, and IFN-γ-deficient mice have demonstrated the importance of nitric oxide [45] and cytokine production [17] for effective vaccination against *Bordetella pertussis*, also being a Gram-negative bacterium, and *Trypanosoma cruzi* [46]. Studies in humans mice and humans have more specifically demonstrated the importance of the cytokine-mediated induction of a mixed Th1/Th17 response for effective vaccination against *Bordetella pertussis* [17,18]. Similarly, a study in chickens has demonstrated that a nasal challenge with *Pasteurella multocida*, a member of the same bacterial family as *Av. paragallinarum*, results in a mixed Th1/Th17 response [47]. Similar to mice, a strong pro-inflammatory cytokine response is likely to be important for effective vaccination in chickens, as demonstrated for a flagellin-adjuvanted vaccine for *Pasteurella multocida* [48]. A nasal challenge with *Av. paragallinarum* in chickens was previously shown to result in the increased local gene expression of *IL-6*, which may contribute to a Th17 adaptive immune response in mammals [49], whereas no changes were observed for the Th1-inducing cytokine *IL-12p35* [50]. In this in vitro study, we found increased gene expression of the Th1/Th17-inducing cytokine *IL-12p40*. We did unexpectedly not observe increased *IL-6* expression by HD11 cells after stimulation with the octavalent vaccine, despite previous studies showing that bacterial stimulation induces *IL-6* expression in HD11 cells [24]. The Th1-incuding cytokine *IL-12p35* and the Th2-incuding cytokine *IL-4* remained undetectable in HD11 cells at any of the time points after stimulation with the octavalent vaccine. The type of adaptive immune response that is triggered by *Av. paragallinarum* during natural infection or vaccination remains interesting for further investigation.

Since the potency of a vaccine does not depend solely on antigen quantity but also on its immunostimulatory capacity, the preservation and consistency of LPS and other PAMPs in inactivated *Av. paragallinarum* vaccines is likely to be important for vaccination [51,52,53,54]. The immunostimulatory capacity of *Av. paragallinarum* bacteria incorporated in the octavalent vaccine could be evaluated using the nitric oxide production assay and was expressed as the half-maximum effective concentration (EC_50_)-value, which here is the vaccine concentration that gives half-maximal nitric oxide production. This opens up the possibility to use the nitric production assay, in addition to antigen quantification methods, to test vaccines containing inactivated *Av. paragallinarum* (and potentially other Gram-negative bacteria) for potency without the use of animals. The bacterial antigens of the octavalent vaccine were extracted from the w/o emulsion with isopropyl myristate, followed by centrifugation steps, to enable the quantification of the immunostimulatory capacity of the vaccine. Testing extracted antigens, rather than the native vaccine formulation, is a strategy that is also used to test inactivated NDV vaccines for potency by an ELISA and has been implemented in the European Pharmacopoeia monograph 0870 [7,9]. Future studies should address the suitability of the nitric oxide assay to discriminate between vaccine batches of different potency, including non-conforming batches, in accordance with the consistency approach [1]. Furthermore, in our study, we observed that high cell passage numbers of the HD11 cell line may slightly affect the sensitivity of the nitric oxide production assay (*not shown*), which needs to be addressed in assay validation studies. The inclusion of reference standards for normalization might be required to improve precision. Finally, the performance of the assay as a QC test should be compared to the animal-based vaccination-challenge test that is currently in place as the gold standard for infectious coryza vaccines [13].

## 5. Conclusions

This explorative study aimed to investigate the immunostimulatory capacity of an inactivated octavalent vaccine for IBV, NDV, EDSV, and *Av. paragallinarum* and to identify immune parameters that could potentially influence vaccine potency. We have found that this inactivated octavalent poultry vaccine activates the chicken macrophage-like cell line HD11 due to the presence of LPS associated with *Av. paragallinarum* antigens, which resulted in the production of nitric oxide and expression of the pro-inflammatory cytokine *IL-1β* and chicken IL-8-like chemokine *CXCLi1*. In contrast, a trivalent vaccine containing inactivated IBV, NDV, and EDSV viral antigens, which are also present in the octavalent vaccine, did not induce nitric oxide production or cytokine expression by HD11 cells, further demonstrating that the responses measured in the assays were specific for the *Av. paragallinarum* antigens of the octavalent vaccine. Furthermore, the nitric oxide production assay was shown to be potentially useful as an in vitro potency test for inactivated poultry vaccines against *Av. paragallinarum* using the EC_50_ of the purified bacteria as a readout for potency. Therefore, this study may contribute to the replacement of current animal-based vaccine QC tests and improve animal welfare.

## Figures and Tables

**Figure 1 vaccines-08-00671-f001:**
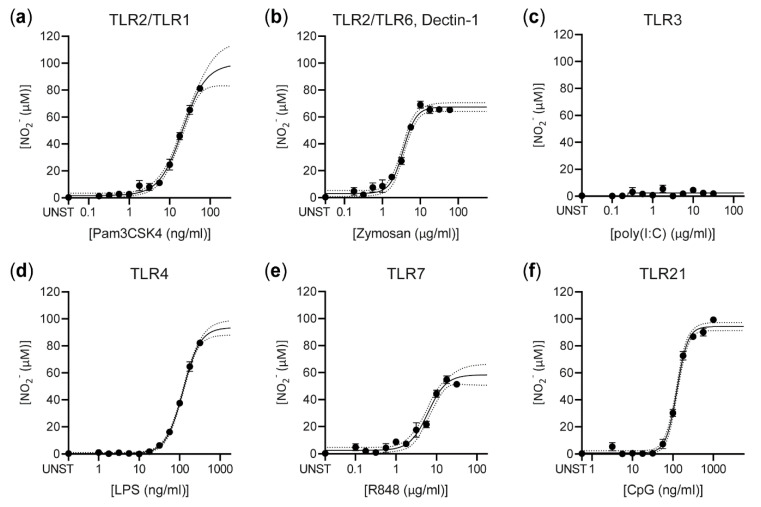
The macrophage-like HD11 cell line produces nitric oxide upon stimulation with a broad range of TLR agonists for 48 h. HD11 cells were stimulated with titrated concentrations of the TLR agonists Pam3CSK4 (**a**), zymosan (**b**), poly(I:C) (**c**), LPS (**d**), R848 (**e**), and CpG (**f**). The expected receptors for each agonist are shown above each panel. Nitric oxide production is expressed as the concentration of nitrite ions (NO_2_^−^) in the cell culture supernatant, as measured by the Griess test. Four parameter logistic curves were plotted together with their confidence intervals (dotted lines). Unstimulated HD11 cells (UNST) were used as a negative control and HD11 cells stimulated with 300 ng/ml LPS (**d**) were used as a positive control in this and all subsequent experiments. The experiment was performed in triplicate. The error bars represent the SEM.

**Figure 2 vaccines-08-00671-f002:**
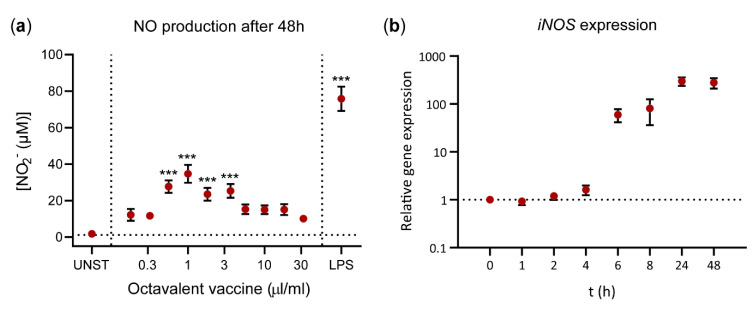
Nitric oxide was produced upon exposure to octavalent vaccine (IBV + NDV + EDSV + 5x *Av**. paragallinarum)* for 48 h. (**a**) HD11 cells were exposed to titrated concentrations of the octavalent vaccine. Moreover, unstimulated HD11 cells (UNST) and HD11 cells stimulated with 300 ng/ml LPS are shown as negative and positive controls. The data comprises three independent technical replicates performed in triplicate. The error bars represent the SEM. A Kruskal–Wallis test combined with Dunn’s multiple comparisons test was used to test for significant induction of nitric oxide production upon stimulation. *** *p* < 0.001. (**b**) RT-qPCR was performed on HD11 cell samples stimulated with 1 µL/mL octavalent vaccine and harvested at the given time points between 0 and 48 h. *iNOS* expression is shown relative to t = 0 h and expressed as 2^-ΔΔCt^ values as calculated using the Livak method and *GAPDH* and *28S* as a reference gene. The experiment was performed in triplicate. The error bars represent the SEM.

**Figure 3 vaccines-08-00671-f003:**
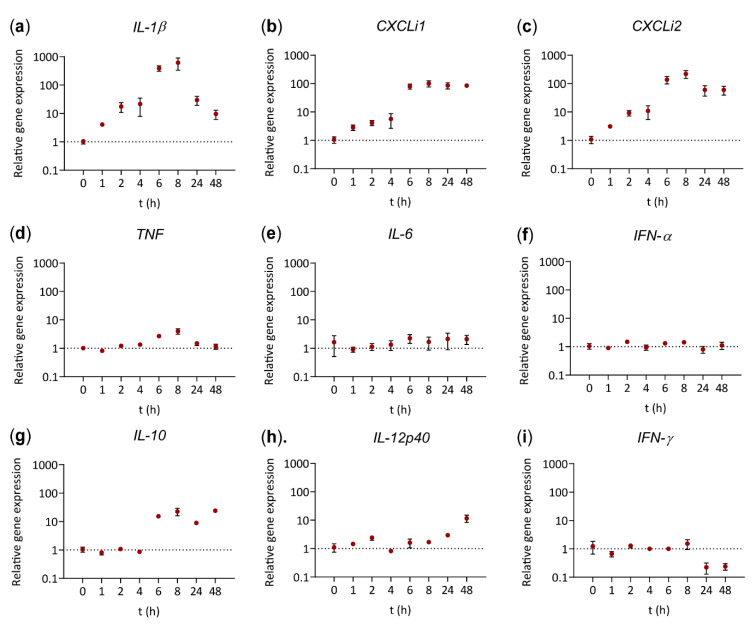
The octavalent vaccine induced the expression of pro-inflammatory cytokines at an early time point, followed by expression of the anti-inflammatory cytokine IL-10 and finally IL-12p40. HD11 cells were stimulated with 1 µL/mL octavalent vaccine and harvested at the indicated time points between 0 and 48 h. RT-qPCR was performed for the cytokines*IL-1β* (**a**), *CXCLi1* (**b**), *CXCLi2* (**c**), *TNF* (**d**), *IL-6* (**e**), *IFN-α* (**f**), *IL-10* (**g**), *IL-12p40* (**h**) and *IFN-γ* (**i**). The relative gene expression levels were normalized against t = 0 h and expressed as 2^-ΔΔCt^ values as calculated using the Livak method and both *GAPDH* and *28S* as reference genes. The experiment was performed in triplicate. The error bars represent the SEM.

**Figure 4 vaccines-08-00671-f004:**
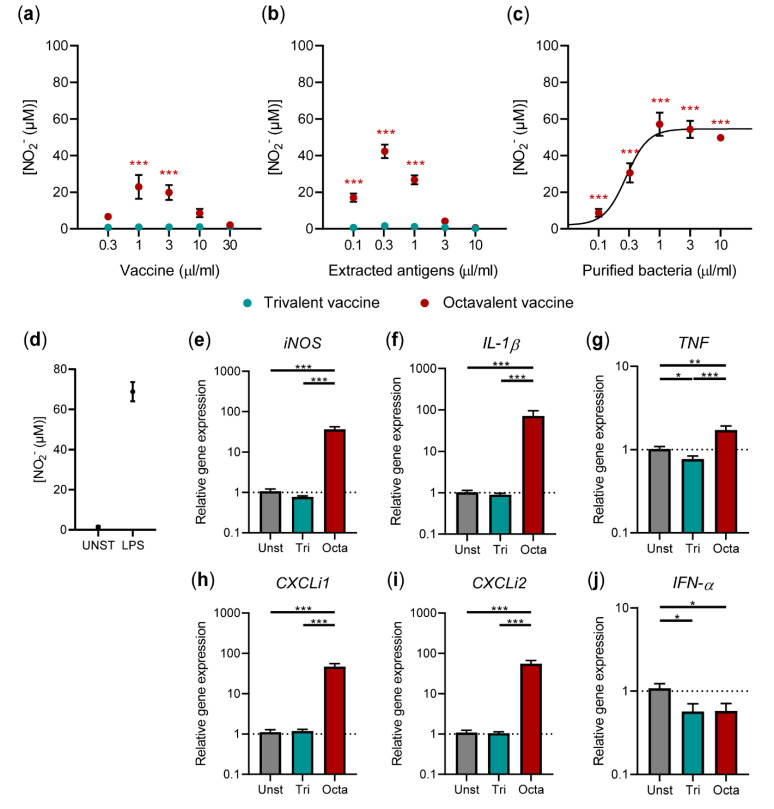
A trivalent vaccine without *Av**. paragallinarum* antigens did not induce nitric oxide production or expression of pro-inflammatory cytokines. (**a**) Nitric oxide production was determined for HD11 cells exposed to titrated doses of trivalent (IBV + NDV + EDSV) and octavalent vaccine (IBV + NDV + EDSV + 5x *Av**. paragallinarum*) for 48 h. (**b**) Antigenic fractions were extracted from the tri- and octavalent vaccines using isopropyl myristate and given to HD11 cells to determine nitric oxide production after 48 h. (**c**) The bacterial pellet was purified from the antigenic fraction of the octavalent vaccine and given to HD11 cells to determine nitric oxide production after 48 h. A four-parameter logistic curve could be calculated and was plotted. (**d**) The controls of the nitric oxide production assay included unstimulated HD11 cells (UNST) and HD11 cells stimulated with 300 ng/ml LPS. (**e**-**j**) Expression levels of *iNOS* (**e**), *IL-1β* (**f**), *TNF* (**g**), *CXCLi1* (**h**), *CXCLi2* (**i**), and *IFN-α* (**j**) by HD11 cells were determined 8 h after stimulation with 1.0 µL/mL tri- or octavalent vaccine. The values are expressed as 2^-ΔΔCt^ values as calculated using the Livak method and both *GAPDH* and *28S* as reference genes. All figures show three independent technical replicates. The error bars represent the SEM. A Kruskal-Wallis test combined with Dunn’s multiple comparisons test was used to test for statistical significance of the data. * *p* < 0.05, ** *p* < 0.01, *** *p* < 0.001. The relative gene expression data were log-transformed prior to the statistical analysis to generate normally distributed data.

**Figure 5 vaccines-08-00671-f005:**
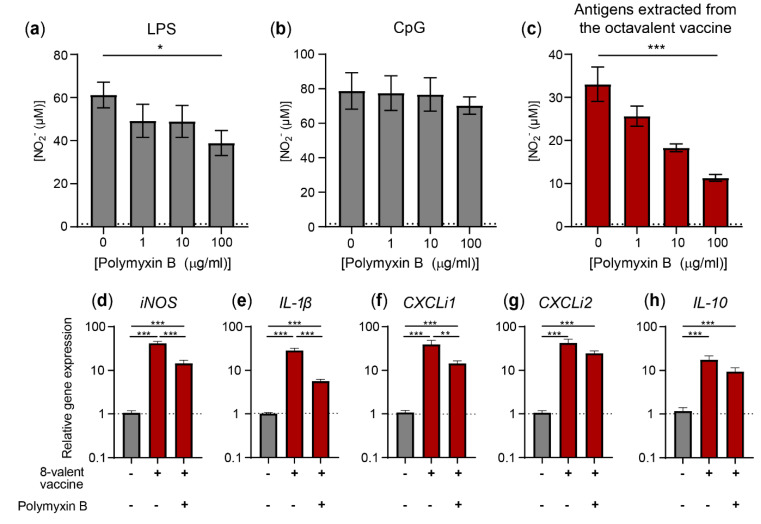
Nitric oxide production upon exposure to the antigenic fraction of the octavalent vaccine could be inhibited by the LPS-binding antibiotic polymyxin B. (**a**–**c**) HD11 cells were pre-incubated in cell culture media without or with 1, 10, or 100 μg/ml polymyxin B for 1 h at 37 °C and subsequently exposed to cell culture media containing 300 ng/ml LPS (**a**), 300 ng/ml CpG (**b**) or 0.5 µL/mL antigens extracted from the octavalent vaccine (**c**) for 48 h. Unstimulated controls are represented by the dotted lines. The first bar of panel **a** is the result of HD11 stimulated with 300 ng/ml without polymyxin B and thus represents the positive control. (**d**–**h**) Expression levels of *iNOS* (**d**), *IL-1β* (**e**), *CXCLi1* (**f**), *CXCLi2* (**g**), and *IL-10* (**h**) by HD11 cells were determined 8 h after stimulation with 1 µL/mL octavalent vaccine and, when indicated, 1 h pre-incubation with 100 µg/ml polymyxin B. The values are expressed as 2^-ΔΔCt^ values as calculated using the Livak method and both *GAPDH* and *28S* as reference genes. The data comprises three independent technical replicates performed in triplicate. Unstimulated controls are represented by the dotted lines. The error bars represent the SEM. A Kruskal–Wallis test combined with Dunn’s multiple comparisons test was used to test for statistical significance of the data. * *p* < 0.05, ** *p* < 0.01, *** *p* < 0.001. The relative gene expression data were log-transformed prior to the statistical analysis to create normally distributed data.

**Table 1 vaccines-08-00671-t001:** Primer and probe sequences. All probes contain the 5′ reporter dye FAM and the 3′ fluorescent quencher TAMRA.

Gene	NCBI Reference	Type	Sequences (5′-3′)
*iNOS*	NM_204961.1	Forward	TGGGTGGAAGCCGAAATA
		Reverse	GTACCAGCCGTTGAAAGGAC
*TNF*	MF000729.1	Forward	CGCTCAGAACGACGTCAA
		Reverse	GTCGTCCACACCAACGAG
*CXCLi1*	NM_205018.1	Forward	CCAGTGCATAGAGACTCATTCCAAA
		Reverse	TGCCATCTTTCAGAGTAGCTATGACT
*GAPDH*	NM_204305.1	Forward	GTGGTGCTAAGCGTGTTATC
		Reverse	GCATGGACAGTGGTCATAAG
*IL-1β*	NM_204524.1	Forward	GCTCTACATGTCGTGTGTGATGAG
		Reverse	TGTCGATGTCCCGCATGA
		Probe	CCACACTGCAGCTGGAGGAAGCC
*IL-4*	NM_001007079.1	Forward	AACATGCGTCAGCTCCTGAAT
		Reverse	TCTGCTAGGAACTTCTCCATTGAA
		Probe	AGCAGCACCTCCCTCAAGGCACC
*IL-6*	NM_204628.1	Forward	GCTCGCCGGCTTCGA
		Reverse	GGTAGGTCTGAAAGGCGAACAG
		Probe	AGGAGAAATGCCTGACGAAGCTCTCCA
*CXCLi2*	NM_205498.1	Forward	GCCCTCCTCCTGGTTTCA
		Reverse	TGGCACCGCAGCTCATT
		Probe	TCTTTACCAGCGTCCTACCTTGCGACA
*IL-10*	NM_001004414.2	Forward	CATGCTGCTGGGCCTGAA
		Reverse	CGTCTCCTTGATCTGCTTGATG
		Probe	CGACGATGCGGCGCTGTCA
*IL-12p35*	NM_213588.1	Forward	TGGCCGCTGCAAACG
		Reverse	ACCTCTTCAAGGGTGCACTCA
		Probe	CCAGCGTCCTCTGCTTCTGCACCTT
*IL-12p40*	NM_213571.1	Forward	TGGGCAAATGATACGGTCAA
		Reverse	CTGAAAAGCTATAAAGAGCCAAGCAAGACGTTCT
		Probe	CAGAGTAGTTCTTTGCCTCACATTTT
*IFN-α*	XM_015277440.2	Forward	GACAGCCAACGCCAAAGC
		Reverse	GTCGCTGCTGTCCAAGCATT
		Probe	CTCAACCGGATCCACCGCTACACC
*IFN-γ*	NM_205149.1	Forward	GTGAAGAAGGTGAAAGATATCATGGA
		Reverse	GCTTTGCGCTGGATTCTCA
		Probe	TGGCCAAGCTCCCGATGAACGA
*28S*	XR_003078040.1	Forward	GGCGAAGCCAGAGGAAACT
		Reverse	GACGACCGATTTGCACGTC
		Probe	AGGACCGCTACGGACCTCCACCA

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
