# Peer review of "Macrophage Activation Assays to Evaluate the Immunostimulatory Capacity of Avibacterium paragallinarum in A Multivalent Poultry Vaccine"

_vaccines, 2020, doi:10.3390/vaccines8040671_

Round 1
Reviewer 1 Report
The aim of the study was to develop a potency test to replace in vivo challenge experiments, a highly relevant and noble goal that will contribute substantially to reducing the number of animals used by the pharmaceutical industry. First the author test the HD11 cell line responsiveness to a variety of TLR ligands. This has been done in tens of other studies but since there are multiple (~3) sublines of HD11 used within the field of avian immunology, it is important information to start with. The authors show that the HD11 line they used has a low to average sensitivity (fig 1 and 3). Then two commercial vaccines containing the same inactivated three viruses (trivalent vaccine) and an octavalent vaccine also containing inactivated multiple serotypes of Av paragallinarum, Gram negative bacteria. The octavalent vaccine induced production of nitric oxide whereas the trivalent vaccine did not and this was caused by the LPS fraction of the octavalent vaccine.
The paper is well written but the major criticism is that it remains unclear what the novelty of the work is. Stimulation of HD11 with TLR ligands or pathogens (live and inactivated viruses and bacteria) have been published in many peer reviewed papers.
In addition after showing that the cell line is not consistent in the production of nitric oxide the authors then omit to include the controls; No production in non treated cells varies significantly and thus for every assay and passage of the cell line a positive control should be included to indicate what the maximum response on that day or of that passage is.
Major comments
Over the decades, the HD11 cell line has gradually changed dependent on the number of passages and it was found that different labs now have cells that show a different sensitivity to LPS. The authors used a line that is less sensitive to LPS, but I am most worried and surprised that their line does not produce IL6 upon stimulation in contrast to most publications [DOI: 10.1016/S0165-2427(01)00260-4;PMID: 27914220;https://doi.org/10.1016/j.dci.2010.03.001;https://edepot.wur.nl/519116]. Please explain the rational to develop a sensitive tool to test a vaccine platform while using a cell line that is not highly sensitive to TLR ligands anymore. It still reacts as shown in fig 1 and 2 but the need for a high dose is a necessity which in my humble opinion is not the most optimal start for development of a tool.
Fig 5a b and c: please explain if the HD11 cells are exposed to polymyxin B for an hour without further stimulation (0ug/ml) the nitric oxide production is on average, 60uM, 80uM and 30uM. Unstimulated cells in fig 2a produce <5uM similar to the values shown in fig 1. If a huge variation is found with polymyxin only the author should include a positive a control to allow variation in responses by the cell line, i.e. include LPS only to obtain the maximum nitric oxide production for that day/that passage number of cells. The decrease in nitric oxide production within an assay is less that the day to day variation. The cell line has been passaged >20x within the time frame of this study which may be one of the reasons for the inconsistent read outs but without a positive control it is unclear of the cells have become “unresponsive”. If the cell line is giving inconsistent data the authors may as well use primary cells that are more relevant and express the PRRs that HD11 is lacking (including multiple C-type lectin receptors). If the cell line is not consistent how can you use the assay to compare vaccine batches or to compare potencies? The paper demonstrates that although they are moving in the right direction this cell line is not suitable (and known to vary) and possible other macrophage like cell lines such as NCSU are more consistent.
It will not come as a surprise that the LPS fraction of Av p is the stimulatory factor in the preparation but the execution to come to this conclusion should be scientifically sound, which isn't if the cell line’s nitric oxide responses vary significantly from day to day or passage to passage. The finding that the LPS fraction/capsule is important for adherence activity was already published in 2015 by Tu in Avian Diseases (59) and reviewed in 2019 by Blackall and whereas Boucher 2014 showing which TLRs and immune genes are regulated upon infection with Av p serovar C3.
Fig 1 HD11 cells are exposed for 48 h to TLR ligands to produce nitric oxide. Exposure of these cells to high levels of ligands induces cell death – please include viability data to ensure the cells are reacting to the ligand and not to high levels of apoptosis. Similar to S1 – for both please include flow plots and show the gating strategy as debris may have been gated out and therefore overestimate the % viability, but that data is missing.
M&M the authors extract the antigenic content of the octavalent vaccine according to ref 7 which is an excellent approach but then they omit to analyse the content of this fraction as was done in ref 7. Please include a more thorough analysis of the material you have used to demonstrate there is more than LPS present.
Minor comments
Please ensure that in every legend you state how long the HD11 cells have been stimulated or it is clear in the M&M but easier for the readers if included in the legend.
Although there is an increase in IL12p40, IL12p35 is not detected and as was shown by Degen et al 2004 JI, heterodimerisation of p35 and p40 is a necessity to yield biological activity. Therefore an increase in p40 only does not have any biological relevance. Please correct and clarify this in the discussion.
Is there a reason why the TNF-a expression is called TNF?
IL6 is a pleiotropic cytokine secreted by a large variety of cells. Since when is IL-6 called a Th17 inducing cytokine? Please refer to papers that demonstrate that in the chicken IL6 production by macrophages leads to induction of Th17 or correct throughout the manuscript.
Fig 4 How do you explain that the trivalent vaccine with adjuvant does not stimulate a macrophage cell line?
Line 387 In Boucher et al 2015 VII the authors conclude that the response to Av p is Th2 driven based on gene expression profiling. Why do the authors suggest a Th1 or Th17 response without taking into account previously published data?
Author Response
Thank you for your feedback and comments. Please see the attachment for our response.

Reviewer 2 Report
This study is well planned and interesting to investigate the immunostimulatory capacity of an inactivated octavalent vaccine for IBV, NDV, EDSV and Avibacterium paragallinarum and to identify various innate immunity immune parameters as potentially influence the vaccine potency. This article has been reviewed well to describe research findings adding very informative technical contents in developing an invitro vaccine potency test. This study is expected to contribute in the replacement of current animal-based vaccine QC tests for improving animal welfare. Please find some specific comments below
1. Please give full name of HD11 cell line.
2. Please review at least two more veterinary/poultry/human bacterial vaccines studies to well justify the role of nitric oxide production and cytokines in induction of vaccine-mediated protection against Avibacterium paragallinarum in chickens (Discussion section).
3. Please give full name of EC-50 in discussion section. Previously it was written as E(C) (that's also fine there). Also describe full name for abbreviated LC-50 word in text.
Author Response
Thank you for your positive feedback and comments. Please see the attachment for our response.

Reviewer 3 Report
The manuscript by Robin et al. presents evidence in support of the capacity of a multivalent vaccine to activate innate immune cells to contribute to vaccine quality testing by in vitro methods.
This is interesting and important work in gathering evidence to support the use of in vitro methodology as a replacement for in vivo methods for determination of virus potency - with potential benefits including improved animal welfare.
My main comment for the authors is that it would be useful to inform the readers as to what the next steps might be before their in vitro testing methods could be formally recognised and adopted by the scientific community? Is their study merely a pilot study to prove the methodology in principle? If so, how would do they intend to follow this work up? Please discuss.
Author Response
Thank you for your positive feedback and comment.
This study was indeed intended as an explorative pilot study, in which we explored the putative use of the macrophage-like cell line HD11 for in vitro potency testing. Doing so, we have identified nitric oxide and pro-inflammatory cytokines IL-1β and CXCLi1 as candidate immune parameters for an in vitro QC test for inactivated poultry vaccines against Avibacterium paragallinarum. However, in this study we have evaluated only one Avibacterium paragallinarum vaccine. As described in the final part of the discussion (lines 425-433), we believe future studies should determine whether the herein described candidate immune parameters can be used for batch-to-batch comparisons and whether these could detect non-conforming batches. Eventually, the ability of the nitric oxide to test vaccines for Avibacterium paragallinarum for potency should be compared to the vaccination-challenge test that is currently in place as a golden standard.
Round 2
Reviewer 1 Report
I would like to thank the authors for their detailed rebuttal.
I have a couple of remarks because what is stated in the rebuttal cannot be found in the revised manuscript.
It was reassuring to read that positive and negative controls were taken along every experiment but in contrast to what is stated in the rebuttal the data are not included. A negative control (dotted line) has been added to some figures but the LPS control is still missing in most figures.
The whole idea behind potency testing in vitro is great but I am still not convinced the authors discussed the drawbacks of this HD11 assay. The assay measuring NO production after stimulation with tri or octavalent vaccine shows that HD11 NO production is not suitable to measure potency of the trivalent vaccine. The octavalent vaccine does induce low levels of NO but the sensitivity is very low - see fig 2A - and it is not clear because different batches of vaccines were not tested if this range in NO production is sufficient to be used for potency testing at commercial level. Therefore the conclusions are not supported by the data. To conclude that this assay is useful for potency testing one should show different batches of vaccines that have been tested in vivo showing differences in potency and then used in vitro to measure NO production and then test for correlation.
I expect that is not possible but please be more critical on your own findings, they are worth publishing but do not overstate the importance of the data if not tested. The assay is not very sensitive and it would be far more useful if the authors come up with suggestions how to improve it.
Reviewer 3 Report
I am happy that the authors have addressed the concerns raised by the reviewers.
Round 3
Reviewer 1 Report
Paper is now acceptable for publication